# Reliability and validity of the PORTRAIT-10 tool for assessing complex health care needs in French-speaking people living with chronic pain

M. Gabrielle Pagé[1,2,3]*, Nicole Tremblay[4], Karen Ghoussoub[2,3], Catherine Hudon[5], Maud-Christine Chouinard[6,7], Manon Choinière[1,3]

**1** Department of Anesthesiology and Pain Medicine, Faculty of Medicine, Université de Montréal, Montreal, Quebec, Canada, **2** Department of Psychology, Faculty of Arts and Sciences, Université de Montréal, Montreal, Quebec, Canada, **3** Research Center, Centre hospitalier de l'Université de Montréal, Montreal, Quebec, Canada, **4** Patient Partner, Research coordination, Réseau universitaire de la santé et des services sociaux, Université de Montréal, Montreal, Quebec, Canada, **5** Department of Family and Emergency Medicine, Faculty of Medicine and Health Sciences, Université de Sherbrooke, Sherbrooke, Quebec, Canada, **6** Faculty of Nursing, Université de Montréal, Montreal, Quebec, Canada, **7** Research center, Centre universitaire de santé et de services sociaux du Nord-de-l'Île-de-Montréal, Montreal, Quebec, Canada

* gabrielle.page@umontreal.ca

## Abstract

Chronic pain (CP) presents multidimensional components, leading individuals to experience complex biopsychosocial needs. However, efficient tools to assess these needs remain scarce. PORTRAIT-10 is a tool designed to measure the complexity of patients' needs. The present study was aimed at documenting the psychometric properties of this tool in a sample of individuals with CP who completed the INTERMED-Self Assessment (IMSA), PORTRAIT-10, Pain Catastrophizing Scale (PCS), and Pain Self-Efficacy Questionnaire (PSEQ). PORTRAIT-10 was re-administered 3 weeks later. The sample comprised 295 participants. Mean age of the respondents was 53.3 ± 9.3 years; 88.3% were females. The median pain duration was 15 years. Results of an exploratory factor analysis showed that a 4-factor solution best fit the PORTRAIT-10 data, with at least 2 of these factors (psychological and social) being consistent with the conceptual framework of the tool. PORTRAIT-10 also showed acceptable internal consistency (Cronbach $\alpha = 0.67$, 0.73) and very good reliability over time ($\rho = 0.85$). Correlation with IMSA was high ($\rho = 0.74$) and as expected, was low with PCS ($\rho = 0.34$) suggesting a very good construct validity. A ROC analysis demonstrated that a PORTRAIT-10 cut-off score of 10 displayed good sensitivity (0.86) and specificity (0.71) in detecting complex care needs in this population. This study provides initial validity and reliability of PORTRAIT-10 and suggests that this tool may be helpful in identifying individuals with CP who have complex needs. Further research is needed to explore the psychometric properties of PORTRAIT-10 in large and more diverse chronic pain populations and to evaluate its impact on clinical outcomes.

**Data availability statement:** Data regarding Portrait-10 is available in supplemental material. Sociodemographic and other self-reported data are not publicly available due to restrictions imposed by the Research Ethics Boards of the Centre hospitalier de l'Université de Montréal given that participants' have not provided consent for data sharing. However, interested parties may contact Research Ethics Boards of the Centre hospitalier de l'Université de Montréal via ethique. recherche.chum@ssss. gouv.qc.ca for data inquiries.

**Funding:** Research grant received from the Ministry of Health and Social Services of the Quebec Government. The funders had no role in the study design, data collection and analysis, decision to publish, or preparation of the manuscript.

## Introduction

Pain is a multidimensional experience encompassing sensory, affective, and evaluative dimensions, with a biopsychosocial approach being essential in the management of chronic pain [1,2]. Many tools are available to measure these dimensions, both for research and clinical purposes. With some exceptions (e.g., McGill Pain Questionnaire), most instruments focus however on one single dimension of the pain and do not consider the various aspects of people's experience that can be simultaneously impacted. Furthermore, the use of unidimensional tools does not inform the need for chronic pain management from a biopsychosocial perspective [3].

Failure to consider the global pain experience often leads to suboptimal treatments, such as over-reliance on pharmacological interventions, which may not address the underlying biopsychosocial factors. This can lead to escalation of pharmacological treatments that are often unsuccessful, a failure which is not necessarily strictly due to the lack of drug efficacy, but rather to the complex biopsychosocial factors people must overcome to access care that meet their global, complex needs [4]. There is some evidence, although heterogeneity is great across studies, that complexity of patients' needs is correlated with some healthcare outcomes [5]. This suggests that better understanding complexity and stratifying patients based on complexity needs might be a clinically-relevant endeavour to pursue.

Pain has historically been conceptualized as a symptom of a disease rather than a disease in and of itself. As a result, specialized care for chronic pain is insufficient to meet the needs of all individuals living with chronic pain both in terms of geographical accessibility and reasonable wait time for patients' first appointment [6–8]. There is thus a need to optimally identify those patients who present with complex needs associated with chronic pain and thus whose care can require access to more integrated care. As such, some tools have been developed to measure this complexity and tailor services/treatments based on complexity of needs and experiences [9]. This type of assessment could also be helpful in assessing patients considered for trials given the limited availability of effective therapeutic interventions in chronic pain. One of these tools, INTERMED, aims to identify patients with low back pain having similar biopsychosocial profiles and to link these subgroups to clinically relevant variables [10,11]. INTERMED was presented as a method to organize coordinated and interdisciplinary health care for patients with complex biopsychosocial care needs. The interest in using such tool for chronic pain lies in the possibility to evaluate various dimensions that influence patient trajectories (biological, psychological, social and health care delivery). These dimensions are considered important factors that dictate the patient needs' level of complexity. For example, among individuals living with rheumatoid arthritis, results showed that a high score on those 4 dimensions was found in approximately 50% of people [12].

While INTERMED is very useful in research, it has important limitations in a clinical practice context: 1) the literacy level required is higher that the recommended Grade 6 reading level and thus inaccessible for many people, 2) it requires a significant amount of time to complete (15−25 minutes), and 3) its scoring system could be too complex for clinicians [9]. In response to these limitations, a new short questionnaire

entitled PORTRAIT-10 was developed (Hudon & Chouinard, https://www.portrait-10.ca). It consists of 10 items with a simplified language and scoring system. The goal of the present study was to examine the factor structure and psychometric properties of PORTRAIT-10 among a sample of individuals living with chronic pain.

## Materials and methods

### Recruitment and participants

Participants were recruited via email from the Quebec Association of Chronic Pain (https://aqdc.info/en/), and inclusion criteria included individuals aged 18 and older with chronic pain lasting more than three months who spoke French and had access to the Internet. The study was approved by the Research Ethics Board of the Centre hospitalier de l'Université de Montréal (22.066-YP). Data collection occurred between February 16th and April 8th 2023, with participants completing online questionnaires on two separate occasions. Informed written consent was obtained from all study participants.

### Procedure

After providing written consent, participants were invited to complete the following online self-reported questionnaires: 1) the reference standard INTERMED-Self Assessment, 2) PORTRAIT-10A, 3) the Pain Catastrophizing Scale, and 4) the Pain Self-Efficacy Questionnaire. The order of administration of the questionnaires was randomized (REDCap Version 13.7.17). Participants in this first round were asked to complete PORTRAIT-10B again 3 weeks later. Participants who responded to both rounds were eligible to win one of 5 prepaid VISA™ cards valued at $100 in a random draw.

### Questionnaires

**INTERMED self assessment.** The INTERMED Self-Assessment (IMSA) is a validated questionnaire which has been designed to assess biopsychosocial complexity and health care needs to optimize care in adult patients [13,14]. The patient's condition is assessed along four dimensions: biological, psychological, social, and health care delivery. Each dimension is evaluated in terms of time: history, current state, and prognosis. The score for each of the four dimensions is obtained by summing answers to 5 questions with answer choices that range from 0 to 3, for a possible subscale score ranging from 0 to 15. The total score is the sum of the 4 dimensions (0–60) [13,14]. Psychometric properties of the IMSA were assessed in a large and heterogenous international sample of adult hospital inpatients and outpatients. The study revealed good reliability and validity of the instrument across different cultures [14]. In the present study, a French version of the IMSA was used, which can be found online and in the user manual [13].

**PORTRAIT-10.** PORTRAIT-10 consists of 10 questions that Hudon and Chouinard proposed in line with INTERMED and their clinical and research expertise with a simplified language and scoring system (Hudon & Chouinard, See S1). Answers to questions are scored from 0 to 4 (Likert type) with a possible total score ranging from 0 to 40. The higher the score, the more complex the patient's needs. Based on the same conceptual framework of the multidimensionality of healthcare needs than INTERMED [15], the following questionnaire structured was used: a) three questions are associated with physical health (general health, pain and medication), b) two questions address psychological health and alcohol/drug use, c) three questions are related to social status (housing, support, and income), and d) two questions ask about the individual's perception of their health needs and the complexity of their clinical condition. As with INTERMED, the aim is to obtain an overall score in order to identify patients with complex needs who might versus might not require access to more integrated care.

**Pain catastrophizing scale.** Pain catastrophizing is described as the tendency to magnify or exaggerate the threat value or seriousness of pain sensations [16–19]. The Pain Catastrophizing Scale (PCS) contains 13 items reflecting the degree to which participants experienced thoughts or feelings when experiencing pain, divided in three sub-scales: rumination, magnification, and helplessness [17]. Each item is scored on a 5-point scale from 0 (not at all) to 4 (all the

time). The PCS has been shown to have adequate to excellent internal consistency (Cronbach α: total PCS = 0.87 to 0.95) [16,20,21]. In the present study, a French version of the PCS was used, called PCS-Canadian French (PCS-CF) [22]. Reliability analyses showed that PCS-CF has a high degree of internal consistency (Cronbach α = 0.91) and test-retest reliability (Pearson r = 0.85) that is comparable to the original PCS [22].

**Pain self-efficacy questionnaire.** The Pain Self-Efficacy Questionnaire (PSEQ) developed by Nicholas in 2007 [23] is a self-administered, one-dimensional questionnaire designed to assess the degree of confidence a person living with chronic pain has to perform different tasks despite the presence of pain. It consists of 10 items; each item is assessed on a Likert-type scale (0 being "not confident at all" and 6 being "completely confident"). Scores are summed to obtain a total score that can ranges from 0 to 60; a high score representing a high degree of self-efficacy. In the present study, the French version of the PSEQ was used [24]. The PSEQ-F was found to be reliable (intraclass correlation coefficient = 0.96); adequate validity with moderate correlations were reported with an abbreviated version of the Disability of the Arm, Shoulder and Hand Questionnaire (r = −0.63) and the Brief Pain Inventory (r = −0.57) [24].

## Statistical analyses

**Descriptive statistics.** Descriptive statistics (e.g., mean, median, frequency table) were used to depict the sociodemographic and clinical characteristics of the participants.

**Exploratory factor analysis.** Exploratory factor analysis (EFA) was used to examine the factor structure of PORTRAIT-10. Data suitability for factor analysis was assessed using the Bartlett's test of sphericity [25] and KMO test [26] with suitability set at > 0.50. Polychoric correlation matrix was used as the input to the EFA models given the ordinal nature of the items [27–29]. As the intent of the analysis was to identify the data's latent structure and considering that the factor structure of PORTRAIT-10 had not been previously examined, EFA was favored over confirmatory factor analysis [28]. The number of factors to retain was determined by using multiple retention criteria—i.e., Empirical Kaiser criterion, Hull method, Kaiser-Guttman criterion, Parallel analysis, Scree plot, Sequential chi-square model tests, RMSEA lower bound, and Akaike Information Criterion, with the function *N_FACTORS* in the EFAtools package in R on the polychoric correlation matrix using maximum likelihood [28]. Theoretical convergence as well as parsimony were also considered. Model fits were also compared across solutions ranging from 1 to 5 factors. Once the number of factors to retain was determined, EFA was performed with the EFA_AVERAGE function of the *EFAtools* package, using the polychoric correlation matrix, a combination of Principal Axis Factoring, Maximum Likelihood, and Unweighted Least Squares as an extraction method, and oblimin rotation. Factors were assumed to be correlated and as such oblique rotation was used [30,31]. The above analyses were carried out with the R statistical package version 4.2.2 [32] and its *EFAtools* [33] and *psych* [34] packages.

**PORTRAIT-10 reliability.** Cronbach alphas (α) were used to assess the internal consistency of PORTRAIT-10A (first administration) and PORTRAIT-10B (administered three weeks later). Spearman correlation coefficients were also calculated to assess test-retest reliability.

**PORTRAIT-10 construct validity.** Construct validity of PORTRAIT-10 was examined using convergent and discriminant validity. Convergent validity was determined by correlating the PORTRAIT-10 scores with the IMSA ones. Discriminant validity was assessed by correlating PORTRAIT-10 scores with those on the PCS and the PSEQ.

**Sensitivity and specificity of PORTRAIT-10.** Receiver operator characteristic (ROC) curves were used to examine possible cut-off scores on PORTRAIT-10 to differentiate people having a complex clinical condition from those who did not. Total score on the PORTRAIT-10 was examined for sensitivity and specificity against the reference point (cut-off score) of 19 on the IMSA (0–18 = no complexity; 19+ = complexity) as the reference value [13,14]. Threshold was determined visually using the ROC curve and by comparing rates of true positive and false positive percentages. Area under the curve (AUC) discrimination measure was also examined as it determines accuracy rate of the questionnaire in discriminating between complex and non-complex cases based on reference point. AUC < 0.70 is considered as poor

discrimination; 0.70 < AUC < 0.80 is considered acceptable discrimination; 0.80 < AUC < 0.90 is considered excellent discrimination, and AUC > 0.90 is considered superior discrimination [35]. Analyses were conducted with the R statistical package and its pROC [36] package.

### Sample size justification

Sample size was estimated for EFA whose focus was to understand the underlying structure of the data and not computing confidence intervals or p-values. There are no clear rules about sample size requirements. Some authors suggest that 300 participants (fewer are required if correlations are high among variables) are sufficient [37], while others suggest a ratio of number of cases to number of variables (e.g., 10–15 participants per item). Recent recommended practices for EFA suggest that sample size depends in part on nature of the data (e.g., high communalities without cross loadings would require a smaller sample size), minimum item loadings, and a number of items per factor [38]. Based on simulation data, a sample size of 200 (a 20:1 subject to item ratio) would provide correct factor structure 70% of the time, and an average error in factor loadings of 0.16 [38].

## Results

### Descriptive statistics

A total of 295 participants were enrolled in the study but not all of them provided demographic information; 88.3% of the respondents were women (197/223); mean age (n = 199) was 53.3 ± 9.3 years and pain duration varied from 2 to 65 years (median (n = 222): 15 years, interquartile range: 17). The vast majority of participants (74.7%) had been living with chronic pain for 10 or more years (<5 years: 10.2%; 5–9 years: 15.1%).

Table 1 shows the response rate for each measurement tool. Incomplete questionnaires were excluded from all the analyses. As shown in Table 1, the average IMSA score is far above the cut-off complexity score (≥ 19/60) [13,14]. In fact, more than 80% of the participants had an IMSA score above the cut-off complexity score of 19/60 suggesting that many people who live with chronic pain commonly continue to have complex needs although their pain has been present for many years.

### Factor structure of PORTRAIT-10

The Barlett's test of sphericity indicated the data was suitable for factor analysis ($X^2(45) = 381.57$, $p < 0.001$), demonstrating that the items on PORTRAIT-10 are sufficiently correlated to warrant factor analysis. The KMO statistic was 0.73, above the minimum standard indicating suitability of the PORTRAIT-10 data for factor analysis. The polychoric correlation matrix is shown in Table 2.

Number of factors to retain in the EFA using various factor retention criteria are listed in Table 3 and the scree plot of eigenvalues is shown in Fig 1. Overall, a 4-factor solution was suggested using the Hull method with Root Mean Square Error of Approximation (RMSEA), Parallel Analysis, and Akaike Information Criterion (AIC).

**Table 1. Participants' response rates (n = 295) for each measurement tool and their average scores.**

|  | Complete questionnaires n % | Median total score (interquartile range) |
|---|---|---|
| **INTERMED-Self Assessment (IMSA)** | 234 79.3 | 28 (11) |
| **PORTRAIT-10A** | 245 83.1 | 20 (7) |
| **PORTRAIT-10B** | 159 53.9 | 19 (6.5) |
| **Pain Catastrophizing Scale (PCS)** | 242 82.0 | 28 (14) |
| **Pain Self-Efficacy Questionnaire (PSEQ)** | 238 80.1 | 2.5 (1.58) |

**Table 2. Polychoric correlation matrix.**

| | Health | Pain | Medi-cation | Mental health | Alcohol/ drug use | Housing | Support | Income | Health needs | Perceived complexity |
|---|---|---|---|---|---|---|---|---|---|---|
| **Health** | 1 | | | | | | | | | |
| **Pain** | 0.34 | 1 | | | | | | | | |
| **Medication** | 0.32 | 0.45 | 1 | | | | | | | |
| **Mental health** | 0.30 | 0.33 | 0.16 | 1 | | | | | | |
| **Alcohol/drug use** | −0.04 | −0.04 | −0.15 | 0.21 | 1 | | | | | |
| **Housing** | 0.17 | 0.14 | 0.09 | 0.36 | 0.11 | 1 | | | | |
| **Support** | 0.11 | 0.09 | −0.05 | 0.25 | 0.02 | 0.42 | 1 | | | |
| **Income** | 0.15 | 0.22 | 0.02 | 0.26 | 0.19 | 0.50 | 0.26 | 1 | | |
| **Health needs** | 0.44 | 0.34 | 0.14 | 0.28 | 0.07 | 0.44 | 0.19 | 0.32 | 1 | |
| **Perceived complexity** | 0.40 | 0.38 | 0.17 | 0.27 | −0.02 | 0.22 | 0.08 | 0.16 | 0.48 | 1 |

To further validate selection of the final EFA solution, an EFA model was ran to compare fit indices for a 1-factor, a 2-factor, a 3-factor, a 4-factor and a 5-factor solution [39]. As shown in Table 4, the 1-factor and 2-factor solutions were not appropriate as the RMSEA values were above 0.06 and the AIC values were high. The 4-factor solution had a better data fit based on RMSEA compared to the 3-factor solution while the 5-factor solution did not generate further improvement.

The 4-factor solution accounted for 51% of the variance (Factor 1: 15%; Factor 2: 15%; Factor 3: 11%; Factor 4: 10%). Factor loadings and communalities ($h^2$) are shown in Table 5. Mean and minimum/maximum loadings for all items across the four factors are displayed in Fig 2. Factor 1 was comprised of items related to biological and global complexity while Factor 2 related to social dimensions of complexity. Factor 3 contained items related to pain and its management. Finally, Factor 4 related to psychological aspects of complexity. It is interesting to note that Factors 2 and 4 are consistent with the conceptual framework of PORTRAIT-10 namely that items 4–5 were designed to measure psychological complexity while items 6–8 were designed to measure aspects of social complexity. Results of the EFA suggest that items measuring global complexity and biological complexity are loading on Factor 1 (with the exception of medication), while the item measuring pain is also loading on Factor 3 measuring biological complexity. Communalities are acceptable for most items (≥ 0.4), but are signaling potential problems for support and income items. Factor intercorrelation coefficients from the oblique rotation solution ranged from 0.11 (between F2 and F3) to 0.19 (between F2 and F4).

**Table 3. Retention criteria used to determine the number of factors in the EFA.**

| Factor Retention Criteria | Number of factors to retain |
|---|---|
| **Empirical Kaiser criterion** | 2 |
| **Hull method with RMSEA** | 4 |
| **Kaiser-Guttman criterion** | 1 |
| **Parallel analysis with EFA** | 4 |
| **Sequential X² model test** | 5 |
| **Lower bound RMSEA** | 3 |
| **Akaike Information Criterion** | 4 |

RMSEA: Root Mean Square Error of Approximation; EFA: Exploratory Factor Analysis.

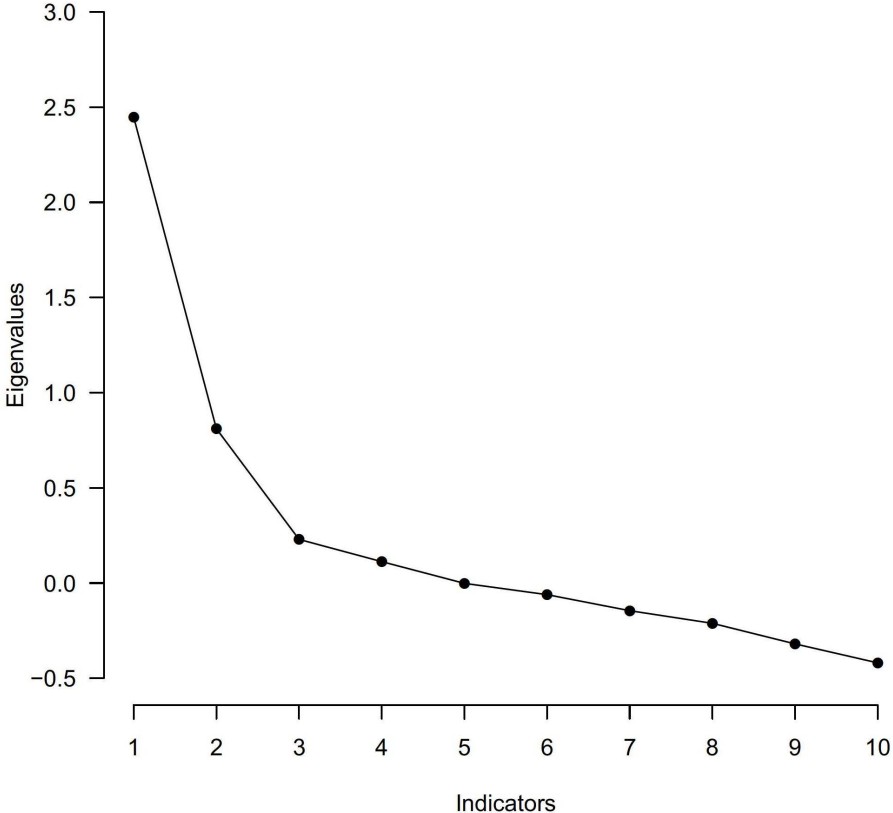

**Fig 1. Scree plot of eigenvalues determined by exploratory factor analysis.**

**Table 4. Comparison between model fits.**

| Fit indices | 1 factor | 2 factors | 3 factors | 4 factors | 5 factors |
|---|---|---|---|---|---|
| Model $X^2$ | 174.96 p<0.001 | 67.37 p=0.46 | 35.58 p=0.148 | 16.21 p=0.264 | 7.24 p=0.303 |
| CFI | 0.81 | 0.94 | 0.98 | 0.99 | 1.00 |
| RMSEA | 0.13 | 0.08 | 0.05 | 0.04 | 0.04 |
| AIC | 104.96 | 15.37 | −0.42 | −5.79 | −2.76 |
| BIC | −17.58 | −75.66 | −63.44 | −44.30 | −20.26 |
| CAF | 0.42 | 0.51 | 0.53 | 0.51 | 0.51 |

CFI: Comparative Fit Index; RMSEA: Root Mean Square Error of Approximation; AIC: Akaike Information Criterion; BIC: Bayesian Information Criterion; CAF: Common Part Account for Index.

## Reliability of PORTRAIT-10

Using Cronbach alpha, PORTRAIT-10A (n=245) and PORTRAIT-10B (n=159) showed acceptable overall internal consistency (α=0.67 and 0.73 respectively). Deletion of any one item did not significantly improve the internal consistency of PORTRAIT-10A (α=0.62–0.70) or PORTRAIT-10B (α=0.66–0.74). Spearman correlation coefficient also showed excellent test-retest reliability of PORTRAIT-10 (ρ=0.85, p<.0001).

**Table 5. EFA factor loadings.**

| PORTRAIT-10 Items | F1 | F2 | F3 | F4 | h² |
|---|---|---|---|---|---|
| **Health** | **0.55** | −0.03 | 0.16 | 0.02 | 0.4 |
| **Pain** | **0.37** | −0.01 | **0.39** | 0.07 | 0.4 |
| **Medication** | −0.04 | 0.04 | **0.91** | −0.05 | 0.8 |
| **Mental health** | 0.14 | 0.16 | 0.19 | **0.57** | 0.7 |
| **Alcohol/drug use** | −0.05 | 0.01 | −0.09 | **0.60** | 0.5 |
| **Housing** | −0.04 | **0.93** | 0.08 | −0.04 | 0.8 |
| **Support** | −0.01 | **0.47** | −0.04 | 0.02 | 0.2 |
| **Income** | 0.09 | **0.49** | 0.01 | 0.07 | 0.3 |
| **Health needs** | **0.69** | 0.22 | −0.10 | −0.04 | 0.6 |
| **Perceived complexity** | **0.72** | −0.04 | −0.05 | −0.01 | 0.5 |

F1: Factor 1; F2: Factor 2; F3: Factor 3; F4: Factor 4; h²: Communalities.

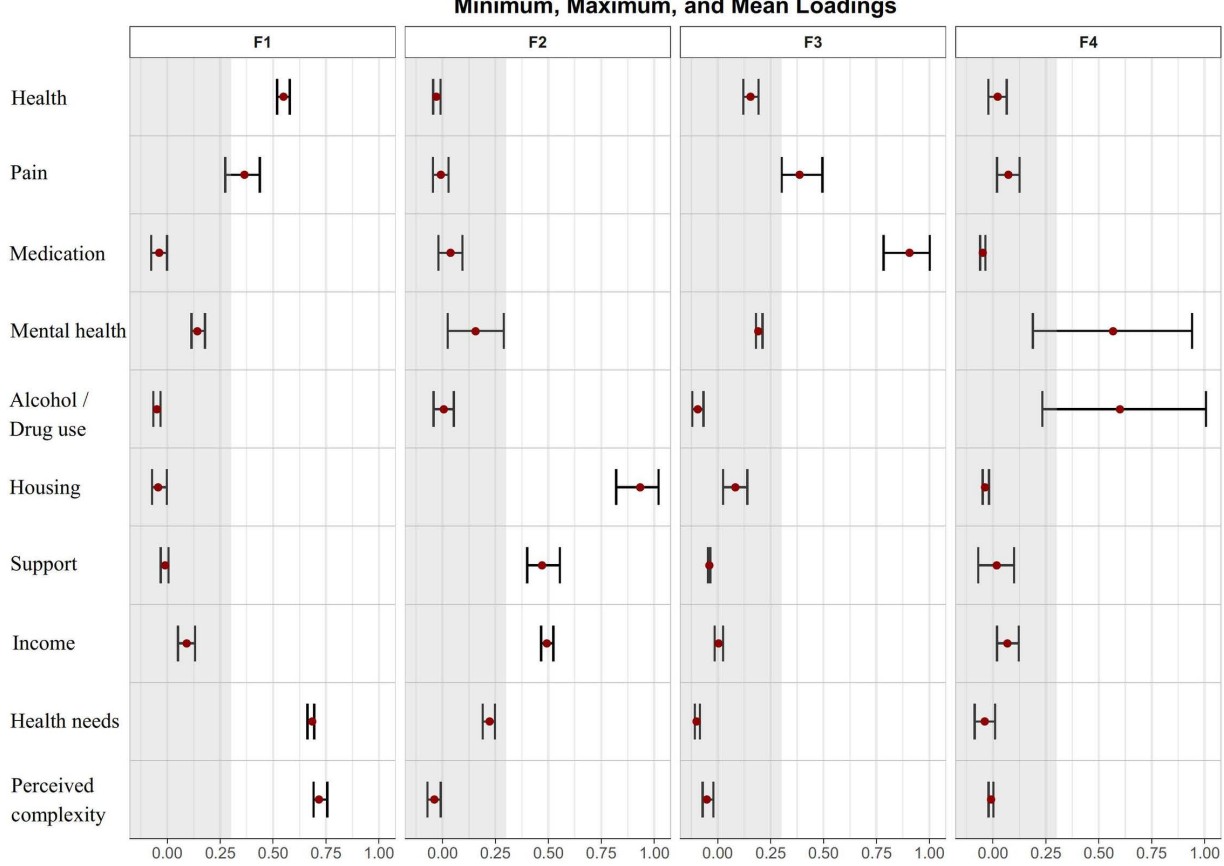

**Fig 2. Factor loadings (mean, minimum and maximum) across all four factors.**

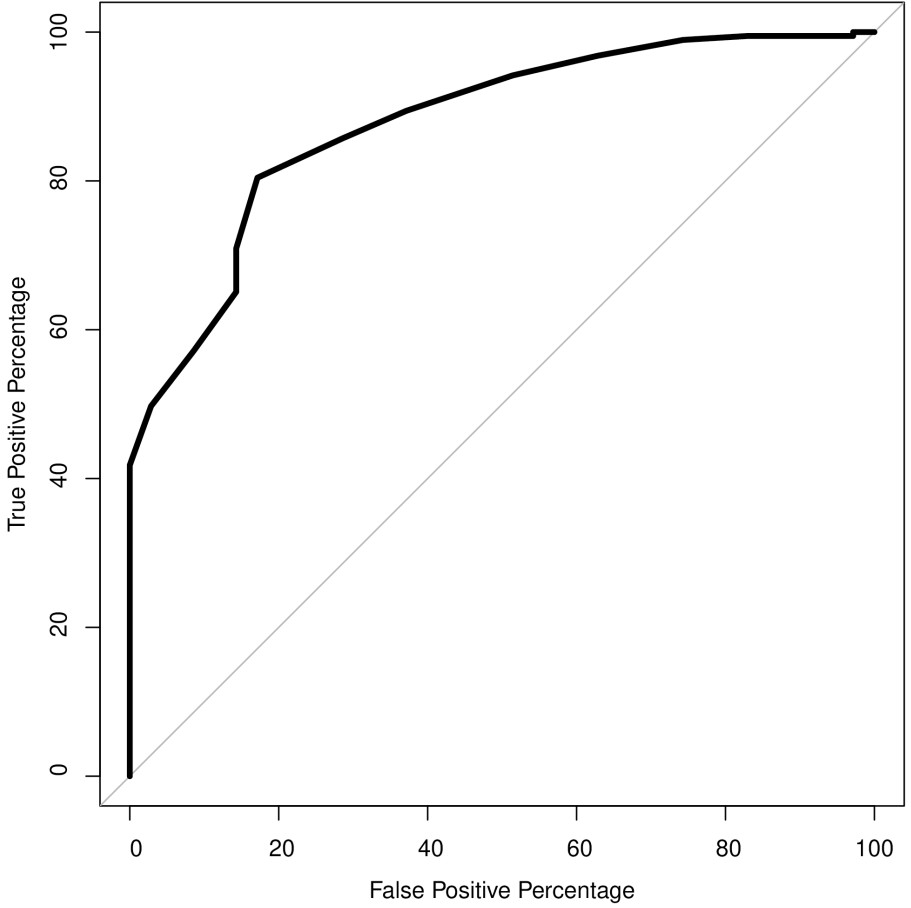

**Fig 3. ROC curve plotting the true positive percentages against the false positive percentages.** ROC area under the curve (AUC) = 0.88.

## Construct validity of PORTRAIT-10

**Convergent validity.** Spearman correlation coefficient showed that PORTRAIT-10 significantly correlated with IMSA ($\rho = 0.74$, $p < 0.0001$) indicating very good convergent validity.

**Discriminant validity.** As expected, PORTRAIT-10 was found to poorly correlate with the Pain Catastrophizing Scale ($\rho = 0.34$) thereby suggesting very good discriminant validity. However, a moderate correlation was found between PORTRAIT-10 and the Pain Self-Efficacy Questionnaire ($\rho = -0.54$, $p < 0.0001$). The higher the scores on PORTRAIT-10, the lower was the perceived pain self-efficacy.

## Sensitivity and specificity of PORTRAIT-10

Results of the ROC analysis are shown in Fig 3 and Table 6. The AUC measure was 0.88, demonstrating excellent discrimination capacities. Based on visual examination of the ROC curve and on the true positive percentages and false positive percentages shown in Table 6, a cut-off score of 10 on PORTRAIT-10 would offer optimal discrimination of complex vs. non-complex cases. This cut-off score had good sensitivity (0.86) and specificity (0.71) meaning that 86% of correct cases were identified—i.e., people were flagged as complex on both the INTERMED Self-Assessment (IMSA) and PORTRAIT-10. However, 28.6% of cases were flagged as complex with PORTRAIT-10 while they were not with the IMSA

**Table 6. True positive percentages and false positive percentages for varying cut-off scores on PORTRAIT-10.**

| PORTRAIT-10 scores | True positive percentages (TPP) | False positive percentages (FPP) | Threshold |
|---|---|---|---|
| 1 | 100 | 100 | -Infinity |
| 2 | 100 | 97.1 | 0.08 |
| 3 | 99.5 | 97.1 | 0.16 |
| 4 | 99.5 | 94.3 | 0.22 |
| 5 | 99.5 | 82.9 | 0.30 |
| 6 | 98.9 | 74.3 | 0.39 |
| 7 | 96.8 | 62.9 | 0.49 |
| 8 | 94.2 | 51.4 | 0.58 |
| 9 | 89.4 | 37.1 | 0.68 |
| 10 | 85.7 | 28.6 | 0.76 |
| 11 | 80.4 | 17.1 | 0.82 |
| 12 | 70.9 | 14.3 | 0.87 |
| 13 | 65.1 | 14.3 | 0.91 |
| 14 | 57.1 | 8.6 | 0.94 |
| 15 | 49.7 | 2.9 | 0.96 |
| 16 | 41.8 | 0.0 | 0.97 |
| 17 | 35.4 | 0.0 | 0.98 |
| 18 | 29.6 | 0.0 | 0.99 |
| 19 | 24.3 | 0.0 | 0.99 |
| 20 | 17.5 | 0.0 | 0.99 |
| 21 | 12.7 | 0.0 | 100 |
| 22 | 7.9 | 0.0 | 100 |
| 23 | 4.8 | 0.0 | 100 |
| 24 | 3.2 | 0.0 | 100 |
| 25 | 1.6 | 0.0 | 100 |
| 26 | 1.1 | 0.0 | 100 |
| 27 | 0.5 | 0.0 | 100 |
| 28 | 0.0 | 0.0 | 100 |

(see Table 6). If the PORTRAIT-10 cut-off score is moved to 12, less true complex cases are detected (70.9%) but only 14.3% of individuals are wrongly flagged as complex.

## Discussion

The goal of this study was to document the factor structure and psychometric properties of a short tool, PORTRAIT-10, designed to assess the complexity of health care needs. The results obtained in a sample of adults living with chronic pain suggest that the instrument is valid and reliable and may be helpful in identifying individuals with complex biopsychosocial needs.

### Psychometric properties

The psychometric properties of PORTRAIT-10 indicate its potential for use in both research and clinical practice. Factor analysis, reliability (internal consistency and test-retest reliability), construct validity (convergent and discriminant validity), sensitivity and specificity were examined in this study to determine the tool's utility in identifying complex health care needs in individuals with chronic pain.

The exploratory factor analysis revealed a 4-factor solution that aligns broadly with the conceptual framework of PORTRAIT-10, although some factors, such as biological and pain-related complexity, do not perfectly correspond with the conceptual model. Two of these factors, psychological and social dimensions, are consistent with the biopsychosocial model of chronic pain. This alignment highlights the validity of PORTRAIT-10 as a measure of multidimensional health care needs. The other two factors, namely the factor 1 that was comprised of items related to biological and global complexity and the factor 3 that was comprised of items related to pain and its management, do not align perfectly with the conceptual framework of INTERMED. However, these two factors account for a non-negligible portion of the variance (15% and 11% respectively), suggesting that these factors capture important aspects of the measured constructs. It is possible that among individuals living with chronic pain, this condition is directly impacting their physical health, but also given its biopsychosocial nature, complexifies the healthcare needs that they have. As such, the distinction between the biological complexity (health, pain, medications) and global complexity (health needs, perceived complexity) becomes more porous. The factors seem interpretable, and the items are loading onto these factors, suggesting that they are measuring coherent constructs [40].

Two items (support and income) had low communalities suggesting that an important amount of variance in these two variables is not explained by the common factors of the EFA. Given that this is a first exploration of PORTRAI-10 in a sample of individuals living with chronic pain and more validation is needed, it could suggest that these two items are not related to the other items, or that the addition of other factors could be considered [38].

The instrument also demonstrated acceptable internal consistency, indicating that its items reliably measure related constructs within each factor. High test-retest reliability was also observed in this study, demonstrating the stability of the tool over time. Despite an attrition rate of 35.1% from the initial assessment to follow-up, the findings suggest that PORTRAIT-10 produces consistent results when applied repeatedly under similar conditions. This reliability is a critical strength, as it supports the use of the tool in clinical monitoring.

PORTRAIT-10 demonstrated strong convergent validity, with a high correlation observed between its scores and those of the INTERMED-Self Assessment (IMSA). This result indicates that PORTRAIT-10 effectively captures the complexity of biopsychosocial needs, comparable to a well-established instrument [10,11]. Its simplicity of administration, short administration time, and easy to understand items make it a good choice of measure to understand complex needs of individuals living with chronic pain.

Discriminant validity was supported by a low correlation between PORTRAIT-10 and the Pain Catastrophizing Scale (PCS), suggesting that PORTRAIT-10 measures a distinct construct. However, the correlation with the Pain Self-Efficacy Questionnaire (PSEQ) was moderate. A possible explanation for this result is that it reflects convergent validity—i.e., individuals with more complex needs are likely to have lower confidence in performing activities while in pain. This finding calls for additional research to clarify the relationship between the complexity of health care needs and pain-related self-efficacy.

Based on ROC analysis, a cut-off score of 10 on PORTRAIT-10 demonstrated good sensitivity (0.86) and specificity (0.71) in distinguishing individuals with complex needs from those without. This means that the tool effectively identifies most individuals with complex needs, although there is room for improvement. Based on these results, a PORTRAIT-10 threshold that is specific to individuals living with chronic pain cannot be suggested. A larger number of individuals with varied clinical profiles would be needed to identify a complexity profile specific to individuals living with chronic pain. Additionally, these sensitivity and specificity findings are preliminary and require replication in other chronic disorders populations. Future research with larger sample sizes and more diverse populations is needed to establish the optimal cut-off score for PORTRAIT-10.

## Clinical implications

The study identified a subgroup of individuals with chronic pain who exhibited particularly complex biopsychosocial needs. The total IMSA scores for these individuals align with findings from previous studies, indicating that a significant proportion

of individuals with chronic pain exceed the threshold for complexity [14,41,42]. Similarly, most participants had total PORTRAIT-10 scores above the complexity threshold, further highlighting the frequent occurrence of complex needs in this population.

Given the multidimensional nature of chronic pain, it is plausible that daily experience of pain fosters complexity in many individuals. The study participants, primarily long-term members of the Quebec Association of Chronic Pain, likely represent a subgroup of individuals who could benefit from individualized, comprehensive services addressing their multi-faceted needs. Using PORTRAIT-10, health care providers could quickly identify individuals with chronic pain who require such services, moving beyond pharmacological pain relief alone, which carries its own risks [43,44].

PORTRAIT-10 offers a quick and efficient way to assess individuals' complexity in clinical settings. Its brevity makes it feasible for routine use in primary care or specialized clinics, allowing providers to screen for complex needs without imposing significant time burdens. Early identification of individuals with high complexity scores can facilitate timely refer-rals to multidisciplinary teams, ensuring that these individuals receive comprehensive care.

By identifying individuals with high levels of psychological and social complexity, PORTRAIT-10 can guide the develop-ment of personalized treatment plans. For example, individuals with high scores on psychological dimensions may benefit from psychotherapy [45], while those with social challenges may require resources to improve social support or access to community services [46]. The tool's high reliability enables clinicians to re-administer it to monitor progress over time and adjust interventions as needed.

Efficiently identifying individuals with complex needs can help optimize resource allocation in health care settings. By focusing intensive interventions on those most likely to benefit, health care providers can potentially reduce unnecessary care and associated costs [47,48].

The multidimensional nature of PORTRAIT-10 aligns with the principles of multidisciplinary care, which is recom-mended for chronic pain management [49,50]. The tool can serve as a common framework for communication among health care providers, facilitating collaboration and ensuring that all aspects of an individual's complex health care needs are addressed.

### Strengths and limitations

This study provides valuable insights into the psychometric properties of a novel questionnaire, PORTRAIT-10. It also explores the possible use of the questionnaire in a clinical setting. However, this study had certain limitations. First, the instrument PORTRAIT-10 was validated in a specific population of people living with chronic pain recruited exclusively online through a patient association, with a high proportion of individuals identifying as women. It is possible that gender, level of education, engagement in patient associations and ease of access to the Internet influence the levels of biopsy-chosocial complexity. More studies are needed to validate PORTRAIT-10 in larger populations of individuals treated in primary care for various chronic disorders. Second, the attrition rate from the initial assessment to the follow-up was high (35.1%) and may have affected statistical power but it is important to point out that with no attrition rate, we might have found even higher correlation coefficients. Third, the present study assessed the validity and reliability of only the French version of PORTRAIT-10. Further research is needed to confirm the psychometric qualities of the English version. Finally, the Cronbach's alphas were moderate. Internal consistency ranged from marginal to acceptable and further evaluation of the scale's psychometric properties is needed.

### Conclusion

This study is the first to examine the validity and reliability of PORTRAIT-10 and the results suggest that the tool can be helpful in identifying individuals with chronic pain who have complex biopsychosocial needs. Further research is needed to document the psychometric properties of PORTRAIT-10 in large samples of individuals living with different chronic clinical conditions including chronic pain. Future studies should also address whether the use of PORTRAIT-10 will lead

to improved care for complex individuals by early detection of their care needs and implementation of adequate treatment strategies.

## Supporting information

**S1 File. Link to the PORTRAIT-10 questionnaire.**
(DOCX)

**S2 File. PORTRAIT-10 data.**
(XLSX)

## Acknowledgments

The authors wish to thank Mrs. Céline Charbonneau and Mr. Pierre Genest who both live with chronic pain and who kindly reviewed the procedures and questionnaires used in this study. Thanks are also due to the Quebec Association of Chronic Pain Patients for the help in recruiting participants. All patients who took part in the present study also deserve to be thanked. Finally, Mr. Marc Dorais from StatSciences Inc. who conducted the initial statistical analyses merits special thanks.

## Author contributions

**Conceptualization:** M. Gabrielle Page, Catherine Hudon, Maud-Christine Chouinard, Manon Choinière.

**Data curation:** M. Gabrielle Page, Nicole Tremblay, Manon Choinière.

**Formal analysis:** M. Gabrielle Page.

**Funding acquisition:** Nicole Tremblay, Manon Choinière.

**Investigation:** Catherine Hudon, Manon Choinière.

**Methodology:** M. Gabrielle Page, Karen Ghoussoub.

**Project administration:** Nicole Tremblay.

**Resources:** Nicole Tremblay.

**Supervision:** Manon Choinière.

**Validation:** Maud-Christine Chouinard.

**Writing – original draft:** M. Gabrielle Page, Karen Ghoussoub, Manon Choinière.

**Writing – review & editing:** Nicole Tremblay, Catherine Hudon, Maud-Christine Chouinard.

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
