## [Decision Letter · Decision Letter 0]

4 May 2025

Dear Dr. Page,

Thank you for submitting your manuscript to PLOS ONE. After careful consideration, we feel that it has merit but does not fully meet PLOS ONE’s publication criteria as it currently stands. Therefore, we invite you to submit a revised version of the manuscript that addresses the points raised during the review process.

Please carefully address the specific points raised by the reviewers, in particular:

Reviewer 1’s comments regarding the clarity of the sample description, relevance to French-speaking populations, and possible applications in trial recruitment.Reviewer 2’s detailed suggestions for improving clarity and precision of the writing. While we invite you to consider these suggestions, please note that the references proposed by Reviewer 2 are authored by the reviewer himself. You are under no obligation to cite them.All comments by Reviewer 3.

We look forward to receiving your revised manuscript.

Kind regards,

Lorenzo Righi

Academic Editor

PLOS ONE

 [Research grant received from the Ministry of Health and Social Services of the Quebec Government]. 

Additional Editor Comments: Please note that I have acted as a reviewer for this manuscript, and you will find my comments below, under Reviewer 3

Reviewers' comments:

Reviewer's Responses to Questions

**Comments to the Author**

1. Is the manuscript technically sound, and do the data support the conclusions?

Reviewer #1: Yes

Reviewer #2: Yes

Reviewer #3: Yes

2. Has the statistical analysis been performed appropriately and rigorously?

Reviewer #1: I Don't Know

Reviewer #2: Yes

Reviewer #3: Yes

3. Have the authors made all data underlying the findings in their manuscript fully available?

Reviewer #1: No

Reviewer #2: Yes

Reviewer #3: Yes

4. Is the manuscript presented in an intelligible fashion and written in standard English?

Reviewer #1: Yes

Reviewer #2: Yes

Reviewer #3: Yes

Reviewer #1: This is an assessment of PORTRAIT-10, a newly developed questionnaire for complex medical needs with the INTERMED-Self Assessment (IMSA), the Pain Catastrophizing Scale (PCS), and the Pain Self-Efficacy Questionnaire (PSEQ) in a French speaking population.

The manuscript is rigorous and well written. It seems to give a thorough analysis of the practical aspects of the questionnaire. I like the way that they have presented their analysis. It is important to note that I am not an expert in this methodology employed but, it seems to follow logical, from what I can see, likely best practice guidelines.

I do not have any general concerns about the article. It might be useful to specify that the article is relevant to the French speaking population in the title.

In the abstract on line 48, it might be nice to see the characteristics (age/pain duration, sex) specific to the respondent population only included - is this 199?

In the introduction, the authors may wish to include that this type of assessment would also have some utility in assessments for patients in trial recruitment, which is key as therapeutic interventions for patients with chronic pain are limited.

Lines 71-75 convey an important message well.

If there is any more information available on the study population this might be useful to understand the generalisability of the analysis.

Perhaps, a little more information on any real-world evidence of improving healthcare outcomes by understanding the health complexity in this way.

Do the authors have any idea why the attrition rate was high, although there are still good numbers of respondents.

Reviewer #2: Title

Current: "Reliability and validity of PORTRAIT-10, a short tool to assess complex health care needs in people living with chronic pain"

Proposed Modification: "Reliability and Validity of the PORTRAIT-10 Tool for Assessing Complex Health Care Needs in Chronic Pain Patients."

Reason: The title is clear, but removing the word "short" would result in a more concise and effective version.

Abstract

Paragraph 1 (Introduction):

Current: "Because of the multidimensional components of chronic pain (CP), individuals often present with complex biopsychosocial needs."

Proposed Modification: "Chronic pain (CP) presents multidimensional components, leading individuals to experience complex biopsychosocial needs. However, efficient tools to assess these needs remain scarce."

Reason: Adding the concept that efficient tools to assess these needs are scarce.

Paragraph 2 (Study Objective):

Current: "PORTRAIT-10 is a short tool designed to measure the complexity of patients’ needs."

Proposed Modification: "PORTRAIT-10 is a tool designed to assess the complexity of patients’ needs."

Reason: Removing the word "short," as it does not add value to the meaning.

Conclusion of the Abstract:

Current: "Further research is needed to document the psychometric properties of the instrument in large samples of individuals living with different chronic conditions including CP."

Proposed Modification: "Further research is needed to explore the psychometric properties of PORTRAIT-10 in larger and more diverse chronic pain populations and to evaluate its impact on clinical outcomes."

Reason: Adding the aspect of clinical impact to guide future research.

Introduction

Paragraph 1:

Current: "Pain is a multidimensional experience comprised of sensory, affective, and evaluative dimensions."

Proposed Modification: "Pain is a multidimensional experience encompassing sensory, affective, and evaluative dimensions, with a biopsychosocial approach being essential in the management of chronic pain."

Reason: Adding a reference to the biopsychosocial approach, which is crucial in chronic pain management.

Paragraph 2:

Current: "Failure to consider the global pain experience can negatively affect its assessment and treatment."

Proposed Modification: "Failure to consider the global pain experience often leads to suboptimal treatments, such as over-reliance on pharmacological interventions, which may not address the underlying biopsychosocial factors."

Reason: Adding the concept of suboptimal treatment, including excessive reliance on medications.

Materials and Methods

Section 2.1 (Recruitment and Participants):

Current: "The study was proposed by email to members of the Quebec Association of Chronic Pain..."

Proposed Modification: "Participants were recruited via email from the Quebec Association of Chronic Pain, and inclusion criteria included individuals aged 18 and older with chronic pain lasting more than three months. Exclusion criteria included..."

Reason: More clarity on selection criteria.

Section 2.2 (Procedure):

Current: "The survey was opened between February 16th and April 8th 2023."

Proposed Modification: "Data collection occurred between February 16th and April 8th 2023, with participants completing online questionnaires on two separate occasions."

Reason: More detail on data collection.

Results

Section 3.2 (Factor Structure of PORTRAIT-10):

Current: "The Bartlett’s test of sphericity was significant (X2(45) = 381.57, p < 0.001)..."

Proposed Modification: "The Bartlett’s test of sphericity indicated the data was suitable for factor analysis (X2(45) = 381.57, p < 0.001), demonstrating that the items on PORTRAIT-10 are sufficiently correlated to warrant factor analysis."

Reason: Adding a clear explanation for non-expert readers.

Discussion

Paragraph 1:

Current: "The exploratory factor analysis revealed a 4-factor solution that aligns with the conceptual framework of PORTRAIT-10."

Proposed Modification: "The exploratory factor analysis revealed a 4-factor solution that aligns broadly with the conceptual framework of PORTRAIT-10, although some factors, such as biological and pain-related complexity, do not perfectly correspond with the conceptual model."

Reason: More precision on the alignment of factors with the conceptual model.

Articles to cite:

Diotaiuti, P., Valente, G., Mancone, S., Grambone, A., Chirico, A., & Lucidi, F. (2022). The use of the Decision Regret Scale in non-clinical contexts. Frontiers in Psychology, 13, 945669. https://doi.org/10.3389/fpsyg.2022.945669

Diotaiuti, P., Corrado, S., Mancone, S., Palombo, M., Rodio, A., Falese, L., Langiano, E., Siqueira, T. C., & Andrade, A. (2022). Both Gender and Agonistic Experience Affect Perceived Pain during the Cold Pressor Test. International Journal of Environmental Research and Public Health, 19(4), 2336. https://doi.org/10.3390/ijerph19042336

Where to cite: The two articles should be cited in the Discussion section to support statements regarding pain experience, complexity assessment, and the role of psychological and social variables. You can cite as follows:

"As demonstrated in previous studies (Diotaiuti et al., 2022), the complexity of pain can be influenced by psychological and social factors, which are essential in understanding the biopsychosocial aspects of chronic pain."

"Furthermore, Diotaiuti et al. (2022) have shown that gender and experience with agonistic tasks can affect perceived pain, which could be relevant for understanding pain perception in the chronic pain population."

Reviewer #3: The study is methodologically sound, and the results support the validity and reliability of the instrument. The topic is highly relevant, and the tool has potential for clinical use.

I list below the points I ask you to consider:

- Better discuss how the characteristics of the sample (online, predominantly female) may limit generalizability of the results.

- Clarify the conceptual interpretation of the four factors, especially the overlap between biological and global complexity.

- Acknowledge that Cronbach’s alpha is moderate. Internal consistency ranged from marginal to acceptable (α = 0.67–0.73), and this should be acknowledged as a limitation.

- Comment on the low communalities for the support and income items and their potential need for refinement.

**Do you want your identity to be public for this peer review?** For information about this choice, including consent withdrawal, please see our Privacy Policy

Reviewer #1: No

Reviewer #2: **Yes: ** Pierluigi Diotaiuti

Reviewer #3: **Yes: ** Lorenzo Righi

---

## [Author Response · Author response to Decision Letter 1]

20 Jul 2025

Response to Reviewers

PONE-D-24-58139

Reliability and validity of PORTRAIT-10, a short tool to assess complex health care needs in people living with chronic pain

PLOS ONE

We thank the reviewers and editor for their encouraging comments about our manuscript. We have addressed these comments below.

EDITOR

Reviewer 1’s comments regarding the clarity of the sample description, relevance to French-speaking populations, and possible applications in trial recruitment.

Response: Thank you for highlighting those comments, and we address them below.

Reviewer 2’s detailed suggestions for improving clarity and precision of the writing. While we invite you to consider these suggestions, please note that the references proposed by Reviewer 2 are authored by the reviewer himself. You are under no obligation to cite them.

Response: We have proof-read the manuscript, and have considered the proposed references. Given that they are not related to the topic of this manuscript, we opted not to include them.

All comments by Reviewer 3.

Response: Please see below for our responses to Reviewer 3’s comments.

Response: We have updated the format of the manuscript.

[Research grant received from the Ministry of Health and Social Services of the Quebec Government].

Response: We have updated this information in the manuscript, and have also included it in the cover letter.

Response: We have consulted our Research Ethics Board. They have finally agreed to let us publish the data regarding the Portrait-10 questionnaire, as this is not sensitive information. Given that we did not seek participants’ consent for data sharing, we are not allowed to publish the sociodemographic data. We have amended the statement and have included the database with the submission:

"Data regarding Portrait-10 is available in supplemental material.

Sociodemographic and other self-reported data are not publicly available due to restrictions imposed by the Research Ethics Boards of the Centre hospitalier de l’Université de Montréal given that participants' have not provided consent for data sharing. However, interested parties may contact Research Ethics Boards of the Centre hospitalier de l’Université de Montréal via ethique.recherche.chum@ssss.gouv.qc.ca for data inquiries."

Response: We have amended the ethics statement accordingly.

Response: We have gone through the list of references, and have not identified retracted manuscripts that are cited.

Additional Editor Comments: Please note that I have acted as a reviewer for this manuscript, and you will find my comments below, under Reviewer 3

Response: Thank you very much.

Reviewers' comments:

Reviewer's Responses to Questions

Comments to the Author

Reviewer #1:

This is an assessment of PORTRAIT-10, a newly developed questionnaire for complex medical needs with the INTERMED-Self Assessment (IMSA), the Pain Catastrophizing Scale (PCS), and the Pain Self-Efficacy Questionnaire (PSEQ) in a French speaking population.

The manuscript is rigorous and well written. It seems to give a thorough analysis of the practical aspects of the questionnaire. I like the way that they have presented their analysis. It is important to note that I am not an expert in this methodology employed but, it seems to follow logical, from what I can see, likely best practice guidelines.

Response: Thank you for your encouraging comments.

I do not have any general concerns about the article. It might be useful to specify that the article is relevant to the French speaking population in the title.

Response: We have modified the title. It now reads: Reliability and Validity of PORTRAIT-10 Tool for Assessing Complex Health Care Needs in French-Speaking People Living with Chronic Pain

In the abstract on line 48, it might be nice to see the characteristics (age/pain duration, sex) specific to the respondent population only included - is this 199?

Response: A total of 295 provided written consent and completed at least part of the questionnaires. We had some participants who did not complete the sociodemographic questionnaire. Numbers in parentheses in the abstract were meant to provide the actual number of participants with available data for each of the variables. We don’t have space in the abstract to provide this clarification (which can be found in the body of the manuscript), and as such have decided to remove those Ns from the abstract.

In the introduction, the authors may wish to include that this type of assessment would also have some utility in assessments for patients in trial recruitment, which is key as therapeutic interventions for patients with chronic pain are limited.

Response: Thank you for this suggestion. We have added the following sentence in the introduction:

“This type of assessment could also be helpful in assessing patients considered for trials given the limited availability of effective therapeutic interventions in chronic pain.”

Lines 71-75 convey an important message well.

Response: Thank you.

If there is any more information available on the study population this might be useful to understand the generalisability of the analysis.

Response: The only other information we have about the study population is the pain duration. Three quarters (74.67%) reported living with chronic pain for 10+ years, while 15,11% and 10,22% reported living with chronic pain for 5-9 years and <5 years, respectively. We have added this information in the introductory paragraph of the results section, as follow:

“The vast majority of participants (74.7%) had been living with chronic pain for 10 or more years (<5 years: 10.2%; 5-9 years: 15.1%).”

Perhaps, a little more information on any real-world evidence of improving healthcare outcomes by understanding the health complexity in this way.

Response: We found one scoping review on this topic. We have added the following information to the second paragraph in the introduction:

“There is some evidence, although heterogeneity is great across studies, that complexity of patients’ needs is correlated with some healthcare outcomes [5]. This suggests that better understanding complexity and stratifying patients based on complexity of needs might be a clinically-relevant endeavour to pursue.”

Do the authors have any idea why the attrition rate was high, although there are still good numbers of respondents.

Response: This is a good question. We hypothesize that the online nature of the study, with no direct contact with the research team, led to high attrition than if the participants had felt more connected with the researchers or the study.

Reviewer #2:

Title

Current: "Reliability and validity of PORTRAIT-10, a short tool to assess complex health care needs in people living with chronic pain"

Proposed Modification: "Reliability and Validity of the PORTRAIT-10 Tool for Assessing Complex Health Care Needs in Chronic Pain Patients."

Reason: The title is clear, but removing the word "short" would result in a more concise and effective version.

Response: We have removed the word ‘short’ as suggested, but kept the last part of the title to make sure we don’t define individuals only as a function of their pain status. The title now reads:

“Reliability and Validity of the PORTRAIT-10 Tool for Assessing Complex Health Care Needs in French-Speaking People Living with Chronic Pain”

Abstract

Paragraph 1 (Introduction):

Current: "Because of the multidimensional components of chronic pain (CP), individuals often present with complex biopsychosocial needs."

Proposed Modification: "Chronic pain (CP) presents multidimensional components, leading individuals to experience complex biopsychosocial needs. However, efficient tools to assess these needs remain scarce."

Reason: Adding the concept that efficient tools to assess these needs are scarce.

Response: Thank you. We have made this modification.

Paragraph 2 (Study Objective):

Current: "PORTRAIT-10 is a short tool designed to measure the complexity of patients’ needs."

Proposed Modification: "PORTRAIT-10 is a tool designed to assess the complexity of patients’ needs."

Reason: Removing the word "short," as it does not add value to the meaning.

Response: Thank you. We have made this modification.

Conclusion of the Abstract:

Current: "Further research is needed to document the psychometric properties of the instrument in large samples of individuals living with different chronic conditions including CP."

Proposed Modification: "Further research is needed to explore the psychometric properties of PORTRAIT-10 in larger and more diverse chronic pain populations and to evaluate its impact on clinical outcomes."

Reason: Adding the aspect of clinical impact to guide future research.

Response: Thank you. We have made this modification.

Introduction

Paragraph 1:

Current: "Pain is a multidimensional experience comprised of sensory, affective, and evaluative dimensions."

Proposed Modification: "Pain is a multidimensional experience encompassing sensory, affective, and evaluative dimensions, with a biopsychosocial approach being essential in the management of chronic pain."

Reason: Adding a reference to the biopsychosocial approach, which is crucial in chronic pain management.

Response: Thank you. We have made this modification.

Paragraph 2:

Current: "Failure to consider the global pain experience can negatively affect its assessment and treatment."

Proposed Modification: "Failure to consider the global pain experience often leads to suboptimal treatments, such as over-reliance on pharmacological interventions, which may not address the underlying biopsychosocial factors."

Reason: Adding the concept of suboptimal treatment, including excessive reliance on medications.

Response: Thank you. We have made this modification.

Materials and Methods

Section 2.1 (Recruitment and Participants):

Current: "The study was proposed by email to members of the Quebec Association of Chronic Pain..."

Proposed Modification: "Participants were recruited via email from the Quebec Association of Chronic Pain, and inclusion criteria included individuals aged 18 and older with chronic pain lasting more than three months. Exclusion criteria included..."

Reason: More clarity on selection criteria.

Response: Thank you. We have made this modification.

Section 2.2 (Procedure):

Current: "The survey was opened between February 16th and April 8th 2023."

Proposed Modification: "Data collection occurred between February 16th and April 8th 2023, with participants completing online questionnaires on two separate occasions."

Reason: More detail on data collection.

Response: Thank you. We have made this modification.

Results

Section 3.2 (Factor Structure of PORTRAIT-10):

Current: "The Bartlett’s test of sphericity was significant (X2(45) = 381.57, p < 0.001)..."

Proposed Modification: "The Bartlett’s test of sphericity indicated the data was suitable for factor analysis (X2(45) = 381.57, p < 0.001), demonstrating that the items on PORTRAIT-10 are sufficiently correlated to warrant factor analysis."

Reason: Adding a clear explanation for non-expert readers.

Response: Thank you. We have made this modification.

Discussion

Paragraph 1:

Current: "The exploratory factor analysis revealed a 4-factor solution that aligns with the conceptual framework of PORTRAIT-10."

Proposed Modification: "The exploratory factor analysis revealed a 4-factor solution that aligns broadly with the conceptual framework of PORTRAIT-10, although some factors, such as biological and pain-related complexity, do not perfectly correspond with the conceptual model."

Reason: More precision on the alignment of factors with the conceptual model.

Response: Thank you. We have made this modification.

Articles to cite:

Diotaiuti, P., Valente, G., Mancone, S., Grambone, A., Chirico, A., & Lucidi, F. (2022). The use of the Decision Regret Scale in non-clinical contexts. Frontiers in Psychology, 13, 945669. https://doi.org/10.3389/fpsyg.2022.945669

Diotaiuti, P., Corrado, S., Mancone, S., Palombo, M., Rodio, A., Falese, L., Langiano, E., Siqueira, T. C., & Andrade, A. (2022). Both Gender and Agonistic Experience Affect Perceived Pain during the Cold Pressor Test. International Journal of Environmental Research and Public Health, 19(4), 2336. https://doi.org/10.3390/ijerph19042336

Where to cite: The two articles should be cited in the Discussion section to support statements regarding pain experience, complexity assessment, and the role of psychological and social variables. You can cite as follows:

"As demonstrated in previous studies (Diotaiuti et al., 2022), the complexity of pain can be influenced by psychological and social factors, which are essential in understanding the biopsychosocial aspects of chronic pain."

"Furthermore, Diotaiuti et al. (2022) have shown that gender and experience with agonistic tasks can affect perceived pain, which could be relevant for understanding pain perception in the chronic pain population."

Response: Thank you for suggesting those interesting reads. We chose not to incorporate the first suggestion, as it discusses a regret scale in a student population, and don’t believe it supports the argument that pain is complex. Actually, the word ‘pain’ was

---

## [Decision Letter · Decision Letter 1]

16 Oct 2025

Reliability and Validity of the PORTRAIT-10 Tppl for Assessing Complex Health Care Needs in French-Speaking People Living with Chronic Pain

PONE-D-24-58139R1

Dear Dr. Gabrielle Page,

We’re pleased to inform you that your manuscript has been judged scientifically suitable for publication and will be formally accepted for publication once it meets all outstanding technical requirements.

Kind regards,

Angelo Marcelo Tusset

Academic Editor

PLOS ONE

Additional Editor Comments (optional):

The authors presented a fully revised version, meeting all the requested corrections and the criteria required for publication of this Journal.

After these considerations, I consider the paper accepted in its current form.

Reviewers' comments:

Reviewer's Responses to Questions

**Comments to the Author**

Reviewer #4: All comments have been addressed

Reviewer #5: All comments have been addressed

2. Is the manuscript technically sound, and do the data support the conclusions?

Reviewer #4: Yes

Reviewer #5: Yes

3. Has the statistical analysis been performed appropriately and rigorously?

Reviewer #4: Yes

Reviewer #5: Yes

4. Have the authors made all data underlying the findings in their manuscript fully available?

Reviewer #4: Yes

Reviewer #5: Yes

5. Is the manuscript presented in an intelligible fashion and written in standard English?

Reviewer #4: Yes

Reviewer #5: Yes

Reviewer #4: The authors studied chronic pain in multidimensional components. They showed PORTRAIT-10 that is a tool designed to measure the complexity of patients’ needs, documenting the psychometric properties of this tool. The manuscript is interesting and well written. Suggestion, "Conclusion" is short and needs to be improved, as well as the resolution of the figures are not good.

Reviewer #5: The authors made the corrections suggested by the reviewers, and the article can be published as is, meeting the publication standards required by the journal.

**Do you want your identity to be public for this peer review?** For information about this choice, including consent withdrawal, please see our Privacy Policy

Reviewer #4: No

Reviewer #5: No

---

## [Editor Report · Acceptance letter]

PONE-D-24-58139R1

PLOS ONE

Dear Dr. Page,

I'm pleased to inform you that your manuscript has been deemed suitable for publication in PLOS ONE. Congratulations! Your manuscript is now being handed over to our production team.

Kind regards,

on behalf of

Professor Angelo Marcelo Tusset

Academic Editor

PLOS ONE